



# Bias correction of daily satellite-based rainfall estimates for hydrologic forecasting in the Upper Zambezi, Africa

Rodrigo Valdés-Pineda[1], Eleonora M.C. Demaría[2,1], Juan B. Valdés[1], Sungwook Wi[3,1], and Aleix Serrat-Capdevila[4,1]

[1] Department of Hydrology and Atmospheric Sciences, University of Arizona, Tucson, 85721, Arizona, USA.
[2] Southwest Watershed Research Center, USDA-ARS, Tucson, 85719, Arizona, USA.
[3] Department of Civil and Environmental Engineering, University of Massachusetts, Amherst, 01003, Massachusetts, USA.
[4] Global Water Practice, The World Bank, Washington, 20433, Washington DC, USA.

*Correspondence to*: Rodrigo Valdés-Pineda (rvaldes@email.arizona.edu)

**Abstract.** The Zambezi Basin is located in the semi-arid region of southern Africa and is one of the largest basins in Africa. The Upper Zambezi River Basin (UZRB) is sparsely gauged (only 11 rain gauges are currently accessible), and real-time rainfall estimates are not readily available. However, Satellite Precipitation Products (SPPs) may complement that information, thereby allowing for improved real-time forecasting of streamflows. In this study, three SPPs for the UZRB are bias-corrected and evaluated for use in real-time forecasting of daily streamflows: (1) CMORPH (Climate Prediction Center's morphing technique), (2) PERSIANN (Precipitation Estimation from Remotely Sensed Information using Artificial Neural Networks), and (3) TRMM-3B42RT (Tropical Rainfall Measuring Mission). Two approaches for bias correction (Quantile Mapping and a Principal Component-based technique) are used to perform Bias Correction (BC) for the daily SPPs; for reference data, the Climate Hazards Group Infrared Precipitation with Stations (CHIRPS) was used. The two BC approaches were evaluated for the period 2001-2016. The bias-corrected SPPs were then used for real-time forecasting of streamflows at Katima Mulilo in the UZRB. Both BC approaches significantly improve the accuracy of the streamflow forecasts in the UZRB.

*Keywords:* Bias Correction, satellite-based rainfall estimates, satellite precipitation products, Upper Zambezi Basin, Streamflows Simulation.





## 1 Introduction

Significant progress has been achieved in recent years in the development and availability of real-time satellite precipitation products (SPPs). However, SPPs still show significant biases that need to be corrected before the rainfall estimates can be used for any hydrologic application such as real-time or seasonal

forecasting. These biases are due to the inaccurate estimation of climate variables and their temporal variations, or the incorrect detection of rainfall events. One example of the latter is the simulation of too many days of low rainfall intensity (< 1 mm), a phenomenon known as the drizzle effect or drizzling (Hay and Clark, 2003; Ines and Hansen, 2006; Christensen et al., 2008; Piani et al., 2010; Ehret et al., 2012; Teutschbein and Seibert, 2012; Chen et al., 2016). These biases can depend significantly on elevation, aspect,

latitude, climate, and rain-producing mechanisms (Demaria and Serrat-Capdevila, 2015); thus it is important to perform location-specific and in some cases season-specific bias corrections.

A study by Kim et al. (2016) indicates that raw satellite-based rainfall estimates require a post-processing of bias correction before data can be useful for forecasting and impact studies. To address this issue, several Bias Correction (BC) methods have been developed: linear scaling (Lenderink et al., 2007),

local intensity scaling (Schmidli et al., 2006), power transformation (Leander and Buishand, 2007), and distribution mapping (Ines et al., 2006; Piani et al., 2010). Other alternative bias correction schemes have been proposed by Kim et al. (2014), Pierce et al. (2015), and Vrac and Friederichs (2015).

These BC methods have been evaluated and compared in a number of studies. For instance, Teutschbein and Seibert (2012) achieved improvement of raw climate variables with all bias correction

approaches mentioned above, and found that capabilities of the BC methods were similar for hydrologic predictions in terms of correcting the mean bias. However, there were clear differences in the ability of these methods to correct standard deviation or percentiles. Fang et al. (2015) found that the power transformation and quantile mapping methods perform equally well in correcting biases for standard deviation and percentiles, whereas the local intensity scaling method performs best in terms of the Nash–Sutcliffe

coefficient and Pearson Correlation Coefficient ($r$).

Various studies dealing with the validation of raw SPPs and the application of BC methods have been conducted in the Zambezi River Basin. For instance, Liechti et al. (2012) compared three daily satellite-based rainfall products (TRMM-3B42, CMORPH, and FEWS) to ground data for the wet seasons of the years 2003 to 2009, and found that TRMM-3B42 is the best product for hydrological modeling in the Zambezi

Basin. Thiemig et al. (2012) compared six satellite-based rainfall estimates against 205 rain gauges distributed over four African basins (Zambezi, Volta, Juba-Shabelle, and Baro-Akobo) for the 2003-2006 period; their findings suggested that African Rainfall Estimation (RFE-2.0) and TRMM-3B42 are the most accurate products. Recently, Gumindoga et al. (2016) evaluated the performance of five BC methods (i.e. linear scaling, elevation bias correction, power transformation, distribution transformation, and quantile mapping)

for the CMORPH satellite-based rainfall estimates in the Zambezi Basin using 54 rain gauges as reference. The authors found that the linear-based BC method successfully corrected the CMORPH estimates of daily mean rainfall. On the other hand, the nonlinear BC schemes (power transformation and quantile mapping)



were most effective in reproducing rainfall totals. Beyer et al. (2016) corrected TRMM-3B42 rainfall estimates over the Upper Zambezi for the period 1998-2010 using the histogram equalization (another name for quantile mapping) as the BC method. The researchers calculated 17 indices describing the characteristics of each rainy season (i.e., duration and rainfall totals of the rainy season, among others) to determine their degree of relationship against maize yields.

In this study we evaluate three SPPs (CMORPH, TRMM 3B42-RT defined here as TMPA and PERSIANN) for the period 03/01/2001 to 04/31/2016; and improve the product qualities using two BC methods: Quantile Mapping Bias Correction (QMBC) and a Principal Components Bias Correction (PCBC) method developed as part of this study. Each of the BC methods is applied to the three SPPs, and the results evaluated for their accuracy in forecasting streamflow for the UZRB. This study seeks to specifically determine: (1) how well the SPPs can represent rainfall estimates for the UZRB, (2) whether SPPs need to be bias-corrected in the UZRB, and (3) whether BC methods improve the rainfall estimates and consequently the hydrologic prediction and forecasts for the UZRB.

## 2 Methods

### 2.1 Study Area

The Zambezi River is the fourth-longest river (~2,574 km) in the continent after the Congo, Nile and Niger (Meier et al., 2011); and it is the longest east-flowing river of Africa. The Zambezi River Basin is located in the semi-arid region of southern Africa (Fig. 1a); the river originates in Zambia and flows through eastern Angola, along the eastern border of Namibia and the northern border of Botswana, along the border between Zambia and Zimbabwe toward Mozambique, and finally drains into the Indian Ocean. While the upper basin is unregulated and hosts the great Barotse Floodplains, the lower basin has two of the largest reservoirs in the world: Kariba Dam and Cahora Bassa Dam. The Zambezi Watercourse Commission is the international body through which the basin states can coordinate joint actions on the river. The basin drainage area is about 1.4 million $Km^2$ and is shared by eight countries (Fig. 1b). Transboundary management of shared water resources is a continuing challenge due to the high spatio-temporal variability of climate within the basin, the increased pressure on the water resources, and the lack of real-time monitoring and predictive capabilities. Consequently, the Zambezi is a very promising and relevant basin to evaluate the performance of hydrologic applications using near real-time SPPs, to support Integrated Water Resources Management (IWRM) (Liechti et al., 2012).

This study focuses on the Upper Zambezi River Basin (UZRB), located in South Africa between the coordinates 10°89′ - 18°98′ S and 18°38′-26°28 E. The drainage area delineated based on the Global Runoff Data Centre (GRDC) gauge at Katima Mulilo (GRDC-1291100) (Fig. 1c) is about 339,521 $km^2$. The historic mean daily flow (1943-2015) at Katima Mulilo stream gauge is 1389.8 $m^3\ s^{-1}$, and the maximum streamflows can reach more than five times the mean flow. The contribution of the UZRB (above Victoria Falls) to the mean annual discharge (~4200 $m^3\ s^{-1}$) measured at the outlet of the Zambezi River is about 25%, being the largest contribution of all tributaries within the basin (Hamududu and Killingtveit, 2016).





Elevation maps obtained from the Hydrosheds Digital Elevation Model (Lehner et al., 2008) show that the UZRB ranges from approximately 938 to 1671 meters above sea level (Fig. 1d). Data on the Upper Zambezi from land cover maps defined at a global scale by Bartholomé et al. (2005) show that the basin is dominated by broadleaved trees (~53%), shrubs (27.3%), and herbaceous plants (16.5%), whereas only a little (~1.5%) of the area is managed or represent agricultural. The spatial distribution of these vegetation types is consistent through the elevational pattern of the basin i.e., broad-leaved forests are located in high-elevation areas (~ > 1000 m.), and shrubs/herbaceous plants are mostly found in low-elevation or flooded areas (Fig. 1e). The slopes within the basin range from flat (floodplains) to moderately steep regions towards the northeast and northwest of the basin (Fig. 1f).

## 2.2 Upper Zambezi River Basin Climatology

To better understand rainfall patterns in the UZRB, the seasonality of the African climate must be described. This seasonality is the result of interactions between atmosphere, ocean, and land as they respond to the annual cycle of insolation (the Earth's seasonal tilt, which makes the area of direct insolation oscillate between the Northern and Southern Hemispheres). This cycle is defined as being the forcing behind the fluctuation of wet-to-dry or warm-to-cold seasons (Giannini et al., 2008). In general terms, seasonal rainfall patterns in the African continent follow a zonally symmetric rainbelt which includes northern Africa during austral winter (Apr-Sep), and southern Africa during austral summer (Oct-Mar), when a deep convection better known as the Inter-Tropical Convergence Zone (ITCZ) moves southwards within the continent (Fig. 2). This rainfall seasonality is extremely important for the continent, because most of Africa depends on the rainy season to supply water for livestock and agriculture (Rockström and Falkenmark, 2015). The seasonal shifts of the ITCZ are also important in controlling part of the West African monsoon, which is a wind system that affects West African regions between latitudes 9° and 20° N. This system is characterized by winds that blow southwesterly during warmer months and northeasterly during cooler months. It is also well known that this monsoon system is driven primarily by sea surface temperature (SST) anomalies and their resulting atmospheric teleconnections, linking oceanic changes with rainfall patterns.

Figure 2 shows that rainfall in the UZRB is strongly seasonal and occurs almost exclusively during austral summer as stated by Meier et al. (2011). The northern part of the basin has mean annual rainfall of about 1100 - 1400 mm yr$^{-1}$ (rain gauge estimates); this declines towards the south, reaching about half of this value towards the southwest. The rain falls in a four- to six-month summer rainy season (see Fig. 3) when the ITCZ moves from the north over the basin between October and March. Evaporation rates are high (1600 mm - 2300 mm) and much water is lost this way in swamps and floodplains, especially in the southwest portion of the basin (Beilfuss and Dos Santos, 2001).




### 2.3 Rainfall and Streamflow Data

Since rain gauges are simply point measurements, it is desirable to have a dense network in order to perform comparisons with satellite-based rainfall estimates (Romilly and Gebremichael, 2011). With this in mind, observed daily rainfall records spanning the period 1998-2013 were obtained from 54 rain gauges

distributed across the Zambezi Basin. From this dataset we were able to extract only 11 rain gauges for the UZRB domain (see Fig. 1c) that had daily records between 2001 and 2013; the amount of records missing from these 11 gauges ranged between 17.9% and 46.7% (Table 1). This high amount of missing daily data combined with the low spatial coverage of rain gauges in the UZRB motivated us to look for other reference gridded rainfall time series, such as those provided by Climate Hazards Group InfraRed Precipitation with

Station data (CHIRPS) (Funk et al., 2015). Rain gauges and CHIRPS estimates were used for comparison of SPPs, but only CHIRPS data were used to bias correct raw SPPs, as detailed in Sect. 2.5.

Table 2 and Figure 4 show daily satellite-based rainfall estimates from 2001–2016 used in this study from three (near) real-time SPPs: CMORPH (the Climate Prediction Center's morphing technique), TRMM-3B42RT (Tropical Rainfall Measuring Mission) defined in this study as TMPA; and PERSIANN

(Precipitation Estimation from Remotely Sensed Information using Artificial Neural Networks. These daily estimates were aggregated from the original three-hourly SPPs and extracted for a squared domain of the UZRB enclosed by 10.5° -19.25° S and 18° - 28° W. This domain at 0.25° of spatial resolution resulted in a total of 1400 grid points for each SPP analyzed in this study.

Daily time series of streamflows at Katima Mulilo stream gauge (Fig. 4) were obtained from the

Global Runoff Data Centre (GRDC) (http://www.bafg.de/GRDC/EN/Home/homepage_node.html). These records were used to calibrate the HYMOD_DS hydrologic model (the model is described in detail in Sect. 2.6), and to evaluate its performance when forced using the raw and bias-corrected SPPs.

### 2.4 Point-to-Pixel and Pixel-to-Pixel Correlations

Every dataset described in Sect. 2.2 was screened by performing point-to-pixel and pixel-to-pixel correlations between each SPP and rain gauges (point-to-pixel), and between each SPP and CHIRPS (pixel-to-pixel). This analysis has been previously performed for the Zambezi Basin by Liechti et al. (2012) and Thiemig et al. (2012) using different groups of datasets. For this study, daily, monthly and yearly temporal scales were analyzed by calculating the Pearson Correlation Coefficient ($r$) between the rain gauge records (or

CHIRPS) and the closest pixel of each SPP as:

$$r\left(x_{i,j}, y_{i,j}\right) = \frac{\sum_{i=1}^{N}(x_{i,j}-\bar{x}_{i,j})\times(y_{i,j,k}-\bar{y}_{i,j,k})}{\sqrt{\sum_{i=1}^{N}(x_{i,j}-\bar{x}_{i,j})^2}\times\sqrt{\sum_{i=1}^{N}(y_{i,j,k}-y_{i,j,k})^2}} \tag{1}$$

where $r\left(x_{i,j}, y_{i,j}\right)$ is the Pearson Correlation coefficient between the time series of rain gauge $x$ (or CHIRPS) at location $i,j$, and the time series of pixel $y$ of satellite $k$ at location $\sim i,j$. The numerator of Eq. (1) $\sum_{i=1}^{N}(x_{i,j} - \bar{x}_{i,j}) \times (y_{i,j,k} - \bar{y}_{i,j,k})$ is the covariance between the time series of rain gauge $x$ at location $i,j$ and pixel $y$ of satellite $k$ at location $\sim i,j$. In the denominator $\sqrt{\sum_{i=1}^{N}(x_{i,j} - \bar{x}_{i,j})^2}$ is the standard deviation for the time series of





rain gauge $x$ at location $i,j$; and $\sqrt{\sum_{i=1}^{N}(y_{i,j,k} - y_{i,j,k})^2}$ is the standard deviation for the time series of pixel $y$ of

satellite $k$ at location $\sim i,j$.

### 2.5 Bias Correction of Daily Satellite-Based Rainfall Estimates

In this study we performed bias correction on raw satellite-based rainfall estimates rather than on temperature data, because rainfall has a more significant influence on streamflow, and because the results of streamflow simulations are consistent with those of corrected rainfall analyses (Fang et al. 2015). Before performing BC calculations, the so-called "drizzle effect" was removed in all SPPs and CHIRPS datasets by replacing daily rainfall accumulation values less than 1 mm with zeros. This is a commonly used approach to

remove the number of drizzle days in raw SPPs estimates (Teutschbein and Seibert, 2012). After this, we assembled the data by month for all 12 months by grouping all daily records of each month i.e., all days of January, all days of February, and so on. This grouping removed seasonality from the raw time series, and therefore potentially improved the efficacy of BC methods.

     All SPPs were bias-corrected utilizing two BC methods: (1) Quantile Mapping Bias Correction

(QMBC); and (2) Principal Components Bias Correction (PCBC); the latter method was proposed by this study. QMBC was selected because recent evaluations by Gumindoga et al. (2016) in the Zambezi Basin concluded that QMBC is most effective in reproducing rainfall totals (the key variable analyzed in this study). PCBC was implemented as an alternative approach, because the method may improve daily rainfall estimates from SPPs by capturing the natural variability of observed rainfall. The CHIRPS dataset was used as the

reference, given its proven accuracy for the African continent and to augment the insufficient spatio-temporal resolution of rain gauges in this study. For instance, CHIRPS has supported effective hydrologic forecasts and trend analyses in southeastern Ethiopia (Funk et al., 2015). A more detailed description of both methods is presented in the following sub-sections.

### 2.5.1 Quantile Mapping Bias Correction (QMBC)

The original QM method is a non-parametric BC method generally applicable to all possible distributions of rainfall (Fang et al., 2015). The QM method applied in this study is based on the initial assumption that both CHIRPS and SPPs distributions are well approximated by the Gamma Probability Density Function (Gamma-PDF). This distribution has been successfully implemented for QM in previous

studies (i.e., Wood et al., 2004; Ines et al., 2006; Piani et al., 2010; Crochemore et al., 2016). The Gamma-PDF used in this study is:

$$\text{Gamma} - \text{PDF}(x_{m,i,j,k}) = \frac{e^{\left(-\frac{x_{m,i,j,k}}{\theta}\right)}x_{m,i,j,k}^{(\lambda-1)}}{\Gamma(\lambda)\theta^\lambda} \tag{2}$$

where $x_{m,i,j,k}$ is the time series of the daily satellite-based rainfall estimates grouped in the month $m = 1{:}12$ (January to December), at the location $i,j$ (1400 grid points for the UZRB, see Sect. 2.3 for details), and for

the SPP $k{=}1{:}3$ (CMORPH, TMPA, and PERSIANN). $\lambda$ and $\theta$ are the respective shape and scale parameters, and $\Gamma(\lambda)$ is the gamma function evaluated at $\lambda$.





The Gamma-PDF defined in Eq. (2) was fitted for CHIRPS and SPPs at every grid-point and for all 12 months separately. The parameters λ and θ were determined using Maximum Likelihood Estimation (MLE).Then using the Gamma Cumulative Distribution Function (Gamma-CDF); each set of parameters was used to calculate the probabilities associated with the daily satellite-based rainfall estimates ($P_{SPP}$). This procedure was also applied for CHIRPS. These probabilities were then used to calculate the corrected rainfall estimates by applying a discrete function of the following form:

$$x'_{m,i,j,k} = \begin{cases} \text{if } x_{m,i,j,k} > 0 \rightarrow x_{m,i,j,k} = F^{-1}(P_{SPP_{m,i,j,k}} | \lambda_{CHIRPS_{m,i,j}}, \theta_{CHIRPS_{m,i,j}}) \\ \text{if } x_{m,i,j,k} = 0 \rightarrow x_{m,i,j,k} = 0 \end{cases} \quad (3)$$

where $x'_{m,i,j,k}$ is the time series of the corrected daily satellite-based rainfall estimates grouped in the month *m*, at the location *i,j*, and for the SPP *k*. The expression $F^{-1}(P_{SPP_{m,i,j,k}} | \lambda_{CHIRPS_{m,i,j}}, \theta_{CHIRPS_{m,i,j}})$ is the Inverse Gamma-CDF evaluated using the daily probability estimated for the month *m*, at the location *i,j* and for the SPP *k*; combined with the shape and scale parameters calculated for CHIRPS ($\lambda_{CHIRPS_{m,i,j}}, \theta_{CHIRPS_{m,i,j}}$), in the month *m*, and location *i,j*

Equation (3) specifies that when a daily rainfall estimate from any SPP is equal to zero, the corrected satellite-based rainfall estimate is also zero. On the other hand, when a daily rainfall estimate from any SPP is larger than zero, the corrected satellite-based rainfall estimate is then calculated using Inverse Gamma-CDF evaluated with the shape and scale parameters of the reference CHIRPS dataset (Fig. 5).

### 2.5.2 Principal Components Bias Correction (PCBC)

Principal Components (PC) are mathematically defined as an orthogonal linear transformation that converts the original data into a new coordinate system. These new variables are uncorrelated linear combinations of the original ones, and are chosen to represent the maximum possible extent of variability contained in the original data (Valdés-Pineda et al., 2016). The most common way to compute PC is by using Singular Value Decomposition (SVD), a method in which any 2-D matrix X can be decomposed into a product of three matrices: two unitary orthogonal matrices U (Principal Components) and V (Empirical Orthogonal Functions) which are known as Eigen or Singular Vectors; and a diagonal matrix S generally known as Eigen or Singular Values. The values of S given in descending order correspond to the amount of variance retained by each PC or Empirical Orthogonal Function (EOF); therefore, the first PC explains the largest amount of variance and then it decreases exponentially towards the last calculated PC. Given this exponential decay of explained variances, PC analyses are commonly used to reduce the dimensionality of large datasets by retaining only a small group of significant components that explain the largest variance (White et al., 1991; Jolliffe, 2002; Hannachi et al., 2007; Valdés-Pineda et al., 2016).

In this study the main goal of using PC was not to reduce the dimensionality of SPPs datasets. Instead the PC analysis was applied as a method to correct the bias of raw SPPs. This alternative proposed approach is named Principal Components Bias Correction (PCBC). The method is applied according to the same rationale used for QMBC; that is, assuming that the statistical properties between reference data and SPPs can be interchangeable as a way to correct raw estimates. To apply PCBC, the original 3-D matrices of CHIRPS and SPPs (space *n* by space *m* by time *k*) can be rearranged as 2-D matrices (space *ij* by time *k*).





After this reshaping is conducted for each dataset, the time series can be standardized along the temporal dimension of each grid-point. For convenience, a covariance matrix ($C$) of standardized data can be calculated either using the spatial domain as $C = X^T X$, or through the temporal domain as $C = XX^T$. This is a common practice used to transform the original matrix $X$ into a new coordinate space, in which the new covariance matrix $C$ is a symmetric square matrix and it is organized along its diagonal.

As a way to compute the corrected values in fewer steps, in this study we used SVD to decompose the original rectangular matrix of CHIRPS and SPPs as: $X_{(n \times m)} = U_{(n \times n)} \times S_{(n \times m)} \times V_{m \times m}^T$  (4), where $X$ is the original raw matrix of CHIRPS and all SPPs (time by space). $U$ is a matrix (time by time) containing the Principal Components of $X$ calculated for CHIRPS and all SPPs. $V$ is a matrix (space by space) containing the EOF of $X$ calculated for CHIRPS and all SPPs. $S$ is a diagonal matrix (time by space) containing the singular values of $X$ calculated for CHIRPS and all SPPs. $T$ is the transpose of matrix $V$. $n$ is the number of days (time) being analyzed, and $m$ is the number of grid-points (space) being analyzed.

After decomposing the matrices of CHIRPS and SPPs (see Fig. 6a and 6b), the bias-corrected daily rainfall estimates are calculated by combining the Eigen vectors computed from the raw SPPs ($U_{SPPs}$ and $V_{SPPs}$), with the singular values calculated for CHIRPS ($S_{CHIRPS}$) (see Fig. 6c). Accordingly, the reconstruction of the matrix containing the bias-corrected daily rainfall estimates is computed as:

$$X'_{(n \times m)} = \underbrace{U_{(n \times n)}}_{SPP_k} \times \underbrace{S_{(n \times m)}}_{CHIRPS} \times \underbrace{V_{m \times m}^T}_{SPP_k} \tag{5}$$

where $X'_{(n \times m)}$ is the matrix containing the bias-corrected daily satellite-based rainfall estimates. $U$ is the matrix containing the PC of $X$ calculated for the $SPP_k$. $V$ is the matrix containing the EOF of $X$ calculated for the $SPP_k$. $S$ is the diagonal matrix containing the singular values of $X$ calculated for CHIRPS.

The matrix reconstruction performed in Eq. (5) uses all singular values calculated for CHIRPS. This means that dimensionality reduction is not applied during PCBC; instead, the total variance contained in the observed data is completely retained and used to correct the raw estimates (Fig. 6c). However, if the objective of the correction is additionally to minimize the noise of the bias-corrected daily rainfall estimates, the retention of a less number of singular values and components (modes of rainfall variability) could eventually improve the performance of this method.

### 2.6 Evaluation of Raw and Bias-Corrected Rainfall Estimates

After applying the methods of bias correction to raw SPPs, a total of six datasets of daily rainfall estimates were created for the UZRB (3 QMBC and 3 PCBC) spanning the period 01/01/2001 to 04/30/2016. These bias-corrected datasets and the raw satellite-based rainfall estimates were compared against CHIRPS estimates by calculating the Bias Percentage as:

$$BIAS(\%) = \frac{\hat{x}_{m,i,j,k} - x_{m,i,j}}{x_{m,i,j}} \times 100 \tag{6}$$

where $\hat{x}_{m,i,j,k}$ are the daily raw or bias-corrected satellite-based rainfall estimates grouped in the month $m$, at the location $i,j$, and for the SPP $k$; and $x_{m,i,j}$ are the daily CHIRPS rainfall estimates for the month $m$, at the location $i,j$.



The bias was initially evaluated by comparing the temporal distributions of daily Bias Percentage for the months of the rainy season (October to March). Then the spatial distribution of mean daily bias in the UZRB was also assessed in order to identify possible connections between bias and the topography of the basin as described in previous studies (see i.e., Gebremichael et al., 2014; Maggioni et al., 2016).

### 2.7 Hydrological Modeling using Raw and Bias-Corrected Data

The bias-corrected datasets and the three original raw datasets (nine in total) were combined with mean daily temperature, obtained from the Princeton Global Forcing Dataset (Sheffield et al., 2006). These daily climate series were used as input forcings to run the distributed version of the HYMOD hydrologic

model defined by Wi et al. (2015) as HYMOD_DS. Additional information about the HYMOD hydrologic model can be found from Moore (1985), Gharari et al. (2013), Remesan et al. (2014), and González-Leiva et al. (2016). Ideally the bias-corrected datasets should be individually used for both model calibration and for simulation purposes as suggested by Serrat-Capdevilla et al. (2014); however, as a way to establish a baseline for comparisons between SPPs, only CHIRPS was used to calibrate HYMOD_DS for the period 2002-2008.

This means that the parameter set of HYMOD_DS calibrated based on CHIRPS was applied to all posterior HYMOD_DS runs forced by raw and bias-corrected SPPs. We consider this to be an acceptable approach due to the fact that all SPPs were bias-corrected using CHIRPS, and because the runs with the raw estimates allowed us to quantify how well the BC methods are in improving real-time streamflows forecasts in the UZRB.

The Genetic Algorithm (GA) introduced by Wang et al. (1991) was used to optimize the 12 parameters of HYMOD_DS. To deal with the large number of model runs required during the GA calibration procedure (in this study 100,000 runs with population size of 1,000 and 100 generations), parallel processing was employed to allow the entire population within a generation to be evaluated at the same time. The Kling-Gupta Efficiency was used as an objective function to evaluate the goodness of the fit between observed and

predicted streamflows (for details see Gupta et al., 2009; Kling et al., 2014). Observed streamflows for the period 2009-2015 were used to validate the HYMOD_DS simulations. All HYMOD_DS runs for calibration were conducted at the Massachusetts Green High Performance Computing Center (MGHPCC) which houses 10,000 high-end computers.

The HYMOD_DS model was setup at 0.25° spatial resolution (for 517 grid cells corresponding to

30 the UZRB above Katima Mulilo) to simulate the basin's hydrologic response (i.e., daily streamflow at Katima Mulilo) for the period 2002-2016 using the nine forcing datasets (raw and bias-corrected rainfall for the three SSPs). These runs resulted in nine time series of daily streamflow simulations at the Katima Mulilo stream gauge. Finally, the ability of both BC methods to accurately reproduce daily observed streamflows was evaluated and discussed to determine the most appropriate dataset for establishing a real-time hydrologic

forecasting system in the UZRB. Figure 7 summarizes the structure of this study.





## 3    Results and Discussion

### 3.1    Climatology and Seasonality of Raw SPPs

The analysis of rainfall seasonality in the UZRB revealed a well-defined difference between wet and dry seasons, suggesting that monthly accumulations in the basin are well captured by CHIRPS and SPPs, and are similarly detected in the rain gauge records (Fig. 8). The overall North-South gradient of rainfall is captured by all the analyzed products, and it is consistent with the climatology of the UZRB described in Sect. 2.2. This gradient, which has been described in previous studies (see i.e. Liechti et al., 2012; Thiemig et al., 2012; and Gumindoga et al., 2016) is the result of the southwards movement of the ITCZ during the austral summer. This movement brings more rainfall to the highlands of the basin. This analysis also revealed that monthly rainfall is overestimated in some months of the wet season; this is likely the result of an overestimation of the number of rainy days over the tropical wet and dry zones of the Zambezi Basin (see Thiemig et al., 2012). Another possible explanation is that SPPs cannot accurately capture the spatial pattern of seasonal rainfall accumulation since they can be also affected by terrain features (see i.e. Gebremichael et al., 2014).

### 3.2    Spatial Distribution of Pixel-to-Pixel and Point-to-Pixel Correlations

Heavy rainfall events and the number of rainy days per year in the Zambezi River Basin are generally subject to small-scale variability (Thiemig et al., 2012); therefore, the validation of SPPs at the pixel scale (the smallest possible spatial scale) offers a significant overview of the spatial convergences and divergences between observed and estimated rainfall. In general, all three SPPs (CMORPH, TMPA, and PERSIANN) show good agreement for daily pixel-to-pixel comparisons against CHIRPS, with $r$ coefficients ranging from 0.52 to 0.83. The areas of agreement and divergence between CHIRPS and SPPS are to some extent consistent with the point-to-pixel correlations calculated between rain gauges and SPPs (Fig. 9a, b, and c). Better results for this analysis were obtained at monthly scales, where pixel-to-pixel correlations within the UZRB reached values ranging between 0.8 and 0.96. This result was also observed for most of the rain gauges, except for one station located near the outlet of the basin (Victoria Falls) in the southwest part of the domain (Fig. 9d, e, and f). This high coherence of monthly results between observed data and SPPs estimates also has been identified by Liechti et al. (2012), who found similar correlation levels between monthly GPCC and CMORPH estimates for the period 2003-2007. On the other hand, a greater variability of pixel-to-pixel correlations was observed at annual timescales, with the lowest levels of correlation over the low plains of the basin. Annual point-to-pixel correlation analyses exhibited poor results, possibly due to the large number of missing records (Fig. 9g, h, and i). At all temporal scales, it can be anticipated that correlations follow a spatial pattern in which larger pixel-to-pixel correlations are found primarily over the highlands of the basin. This means that the ability of SPPs to represent the daily, monthly, and annual rainfall totals is better over the mountainous areas of the UZRB. This pattern is observed for CMORPH and TMPA products but not for PERSIANN which follows a more homogeneous pattern (see Fig. 7). Similar results for this analysis have been also found by Dinku et al. (2007); Romilly and Gebremichael (2011); and Thiemig et al. (2012).





### 3.3    Evaluation of Daily Bias from Raw and Bias-Corrected SPPs

Both Quantile Mapping (QMBC) and Principal Components (PCBC) Bias Correction methods significantly improved CMORPH, TMPA, and PERSIANN daily rainfall estimates over the UZRB. For instance, scatter plots for all months of the rainy season (October to March) revealed significant deviations of raw estimations with respect to the 1:1 line (Fig. 10). In general, it can be observed that PERSIANN (infrared-based product) tends to overestimate daily rainfall in the UZRB. This overestimation occurs during the rainy season and is probably due to the lack of calibration against ground observations (Asadullah et al., 2008; Thiemig et al., 2012). The raw CMORPH product also overestimates daily rainfall, but to a lesser extent than PERSIANN. This finding is in agreement with Yang and Luo (2014), whose analyses of the distribution of biases concluded that CMORPH and PERSIANN overestimated daily rainfall in the arid region of northwest China.

On the other hand, the TMPA product outperformed, generating the best estimates for the UZRB. This could be attributed to the integration of a larger number of sensors used in the calibration of the TRMM 3B42-RT product (see details in Maggionni et al., 2016), or to the quantity of ground data used in the historical adjustment of the different SPPs algorithms. Similar results have been stated by Romilly and Gebremichael (2011), who found that the microwave-based products TMPA and CMORPH outperformed the infrared-based product PERSIANN for Ethiopian basins. Despite these findings it is not possible to assure that SPPs will always perform spatially as described above, because the scarcity of rain gauges can adversely affect the historical bias adjustment of the TMPA algorithm; and because SPPs such as CMORPH and PERSIANN are designed more to estimate tropical convection rain patterns than for isolated convection systems in arid or semiarid regions (Dinku et al., 2010a).

### 3.4    Temporal Distribution Daily Bias

As discussed in Sect. 3.3, raw PERSIANN and CMORPH estimates revealed positive daily bias during the rainy season (October to March) (see Fig. 11a and c). On the other hand, the bias calculated from the raw TMPA product turned out to be less variable and closer to zero for all months under analysis (Fig. 11b), confirming its greater capacity to estimate daily rainfall in the UZRB (see also Liechti et al., 2012; Thiemig et al., 2012). Both BC methods reduced the biases; the corrections attained by applying QMBC revealed less variability than those corrections resulting from PCBC. This analysis suggests that the statistical QMBC method is slightly better in reducing biases than PCBC, because it better corrects the statistical properties of rainfall accumulation while also providing narrower variability ranges. However, it is important to indicate that the larger variability observed for the bias calculated from the rainfall estimates corrected by PCBC, can hypothetically be minimized if a less number of singular values and principal components are retained during the reconstruction stage. This assumption must be tested in future research in order to know how the performance of this method can be enhanced to obtain better rainfall estimates.





### 3.5 Spatial Distribution of Daily Bias

Both BC methods greatly reduced positive bias, but in some cases they simultaneously increased negative bias to a smaller extent. The net effect of this is bias reduction that is further skewed toward underestimation of rainfall. The spatial pattern of mean daily bias calculated using raw and corrected datasets
revealed that the PCBC method is able to retain the physical characteristics of the bias-corrected rainfall estimates by reducing bias throughout the spatial structure of SPPs (Fig. 12, a, c ,d, f, g, and i). This differs from QMBC which is instead applied to each grid point independently (see Fig. 12b, e, and h). A bimodal pattern of bias in the UZRB is evident and consistent to some extent with the topography of the basin. For instance, in the lowlands located in the mid-UZRB, CMORPH and TMPA (raw and bias-corrected data)
exhibit a marked pattern of negative bias (underestimation), while PERSIANN data show a still evident but slightly positive bias (overestimation) (see Fig. 12a, c, d, f, g, and i). This finding diverges from Romilly and Gebremichael (2011). They found for Ethiopian basins that TMPA (3B42RT) and CMORPH tend to overestimate rainfall at low elevations but give reasonably accurate results at high elevations, whereas PERSIANN gives reasonably accurate values at low elevations but underestimates at high elevations. This
difference may be due to the fact that the two studies were conducted at different elevational ranges, since Ethiopian basins can be as high as 4500 m.a.s.l. but the UZRB reaches only to about 1600 m.a.s.l. (Gumindoga et al., 2016). However, the role of other factors like orography and aspect within the basin could be more important than we actually know, and therefore further research must be carried out to determine the influence of these factors on the satellite-based rainfall estimates. On the other hand, it is well-known that the
performance of SPPs over tropical or equatorial regions and the semiarid and mountainous regions of Africa is completely different (Haile et al., 2013; Diem et al., 2014); therefore, it is also possible to hypothesize that the differences observed for SPPs in the UZRB could be dependent of the transition from humid subtropical to warm semiarid climate, which eventually could reduce the capacity of SPPs to capture the North-South rainfall gradient.

### 3.6 Hydrological Sensitivity of Raw and BC SPPs

As expected, raw overestimations of daily rainfall observed for CMORPH and PERSIANN resulted in overestimations of streamflows at Katima Mulilo stream gauge (Fig. 13a and c). Raw TMPA estimates also resulted in overestimation of streamflows, but to a much lesser extent compared to the other two raw SPPs
(Fig. 13b). In fact, applying the correction to TMPA significantly improved the ability to forecast daily streamflows, with the results from PCBC slightly better than those from QMBC (see Table 3 and Fig. 13e and h). The correction applied to CMORPH and PERSIANN datasets also resulted in significant improvements in the ability to forecast daily streamflows; however, the QMBC correction outperformed the PCBC, suggesting that the latter approach reduces the benefits of the correction when data are poorly cross-correlated (Fig. 13a,
35   d, g). These findings indicate that bias correction methods not only improve the quality of corrected SPPs, but also have a direct influence on hydrological simulations in the UZRB. This result is in agreement with previous studies as those carried out by Teutschbein and Seibert (2012), Casse et al. (2015), and Crochemore





et al. (2016) who also emphasize on the importance of applying bias correction on raw satellite-based rainfall estimates as a way to improve streamflow forecasts.

## 4    Conclusions

Many studies have been conducted during the last decade to determine the impact of past, present, and future climate over the spatio-temporal availability of water resources at the catchment scale. In this regard, SPPs are a valuable source of information, particularly in sparsely gauged regions like the UZRB. In this study, we bias-corrected and evaluated the accuracy and sensitivity of three SPPs – CMORPH, PERSIANN, and TRMM 3B42-RT (TMPA) – for real-time hydrological applications in the UZRB. In this regard we conclude that seasonality of rainfall in the UZRB is well captured by CHIRPS and all three SPPs, and it is also detected in the rain gauge records. However, during the wet season, the monthly rainfall is overestimated for some months; this is probably due to overestimation in the number and/or amount of rainy days and perhaps because SPPs cannot adequately capture the spatial pattern of rainfall due to climate or landscape variations within the UZRB.

Good relationships (correlations) were observed for the pixel-to-pixel (point-to-pixel) comparisons between CHIRPS (rain gauges) data and raw SPPs. The best correlations between CHIRPS and SPPs always occur at monthly timescales, suggesting that the cyclic pattern of rainfall is well represented for these timescales. Additionally, at all temporal scales the greater correlations for CMORPH and TMPA were mostly located in the basin highlands, meaning that the predictive ability of SPPs is greater over the mountainous areas. However, given the low elevational range of the basin, it is premature to argue that this pattern is exclusively related to elevation or topography. More specific analyses will be required to determine the real influence of the landscape features i.e. elevation, topography, aspect, or latitude of the basin on the raw SPPs estimates.

In relation to raw rainfall estimates we found that the TMPA product outperformed PERSIANN and CMORPH, because the former produces more realistic raw estimates for the UZRB. In general, PERSIANN and CMORPH have a tendency to overestimate daily rainfall in the basin, with the former deviating more positively from observed data than the latter (positive bias). These differences can be attributed to the integration of a larger number of sensors used in the calibration of the TMPA product, as well as the quantity of historical ground data used in the historical adjustment of the different SPPs algorithms. They could also be attributed to the way the algorithms use the information obtained by the different sensors to represent rainfall events i.e., SPPs cannot adequately discriminate between stratiform and convective rainfall events.

Both BC methods (Quantile Mapping and Principal Components) satisfactorily improved daily raw SPPs estimates in the UZRB. The QMBC method seems to be slightly better than PCBC, since it can better correct the statistical properties of rainfall accumulation and thus provide narrowest intervals of variability. However, the retention of a less number of components (and singular values) in the reconstruction stage of PCBC could eventually resolve this issue, but further research will be needed to verify how this assumption can affect the performance of this method.



The bias correction achieved significantly improved the ability of these products to forecast streamflows at a daily scale. This result is in agreement with previous findings that have suggested that rainfall correction methods have more significant influence than temperature correction methods. PCBC performed better than QMBC only when the estimates in the datasets were highly cross-correlated. Both BC methods improved the quality of SPPs, and consequently they have a direct influence over the accuracy of hydrological simulations.

Improvements in hydrological forecasts obtained by bias-correcting raw rainfall estimates can help to enhance the operation of reservoirs, planning for irrigation, and construction of hydraulic works, among other things. This process is undoubtedly relevant for forecasting future scenarios in which the pressure exerted by users of water resources increases. Additionally, it is worth mentioning that since these BC methods are assumed to be stationary, the correction algorithm and its parameterization can be valid for current climate conditions. However, further research is necessary on this topic to clearly determine the ability of these BC methods to correct future raw estimates in the UZRB, under different climate scenarios.

## 5 Competing interests

The authors declare that they have no conflict of interest.

## 6 Acknowledgments

This research was supported by NASA-USAID SERVIR Program (Award 11-SERVIR11-58). The supercomputing facilities managed by the Research Computing Department at the University of Massachusetts provided the calibration of HYMOD_DS. Webster Gumindoga provided all the rain gauge records used in this study, and Chris Funk from the USGS provided valuable comments on the CHIRPS dataset. All these contributions are gratefully acknowledged.

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





**Figures**

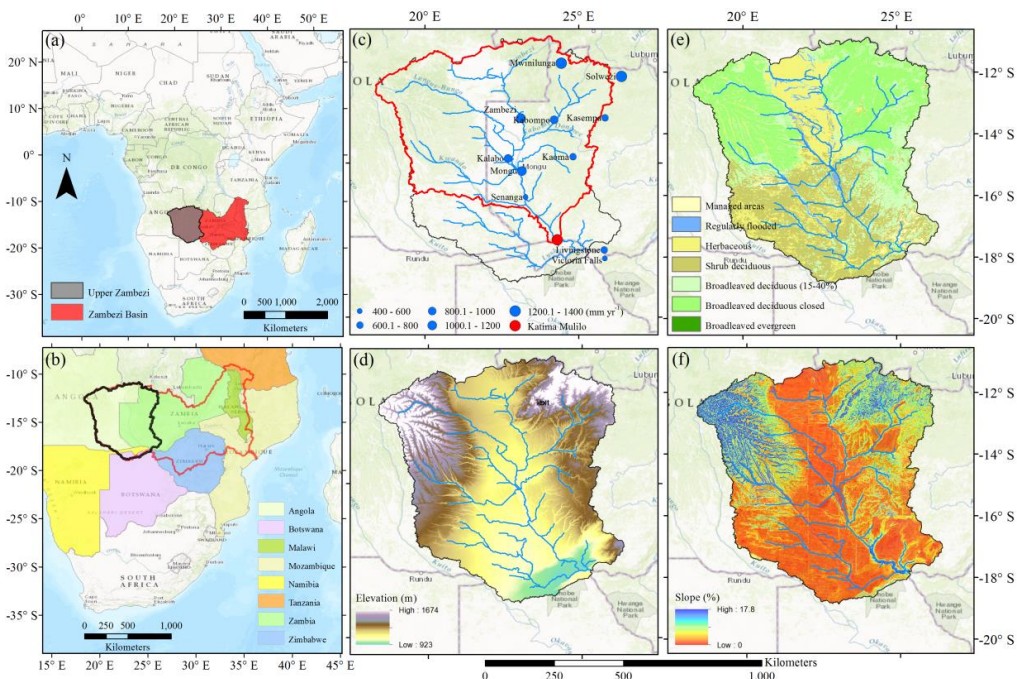

**Figure 1. (a) The Zambezi River Basin in the African continent; (b) Transnational overview of the (Upper) Zambezi Basin; (c) Upper Zambezi, showing its river Network, rain gauges (blue), and Katima Mulilo stream gauge (red) and delineated basin; (d) Digital Elevation Model (DEM) of the basin based on Hydrosheds (90 meters resolution) (Lehner et al., 2008); (e) land cover in the Upper Zambezi based on Bartholomé et al. (2005); (f) slopes map for the Upper Zambezi.**





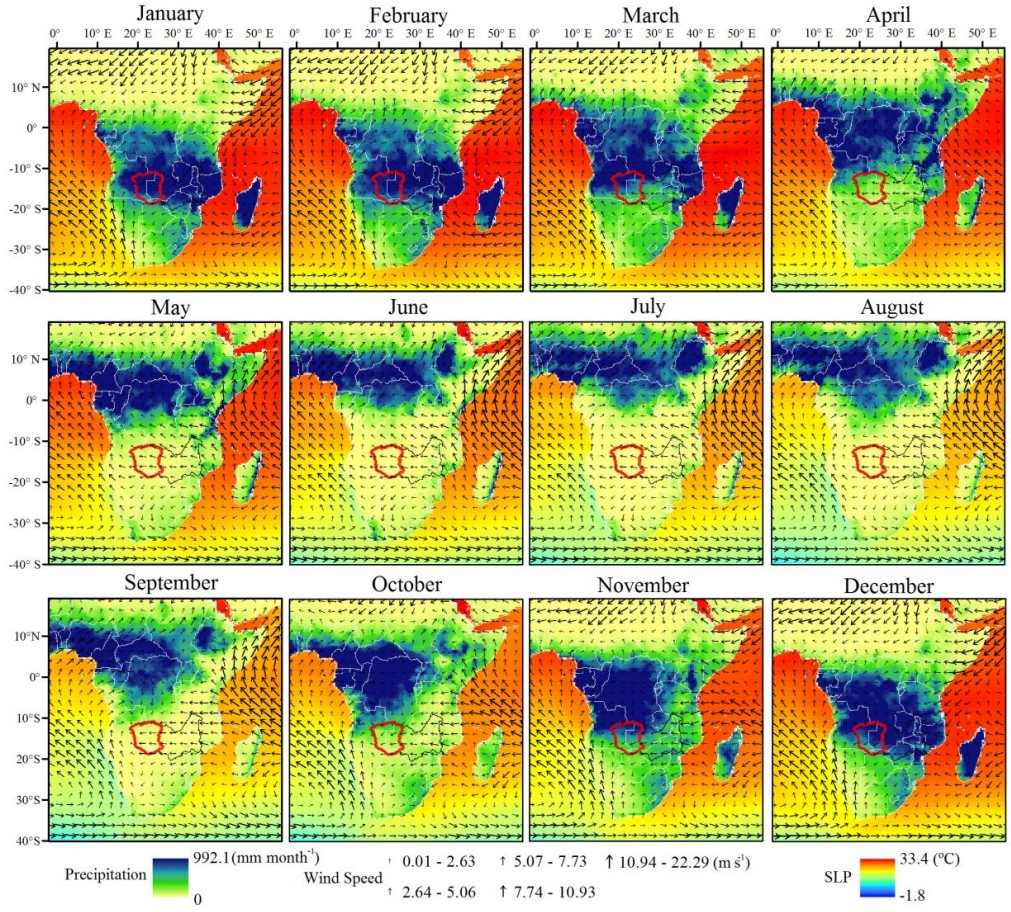

**Figure 2. Delaware long-term mean monthly precipitation for the period 1948–2014 (Matsuura and Willmott, 2012). Maps are superimposed with wind arrows (925 hPa, 1000 m.a.s.l.) and SST for the same period obtained from NCEP/NCAR Reanalysis.**




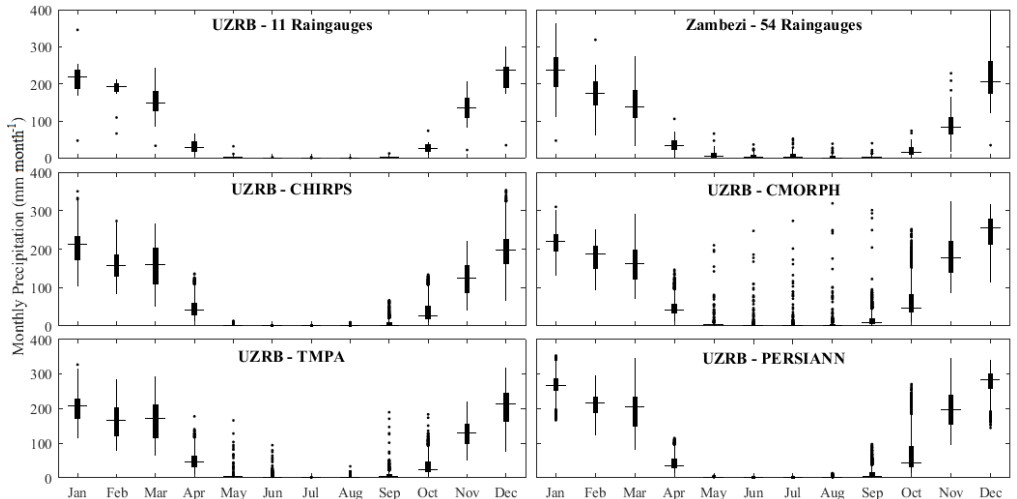

**Figure 3. Mean Monthly Precipitation calculated from 11 rain gauges located in the UZRB, 54 rain gauges located in the Zambezi Basin, CHIRPS, CMORPH, TMPA, and PERSIANN.**

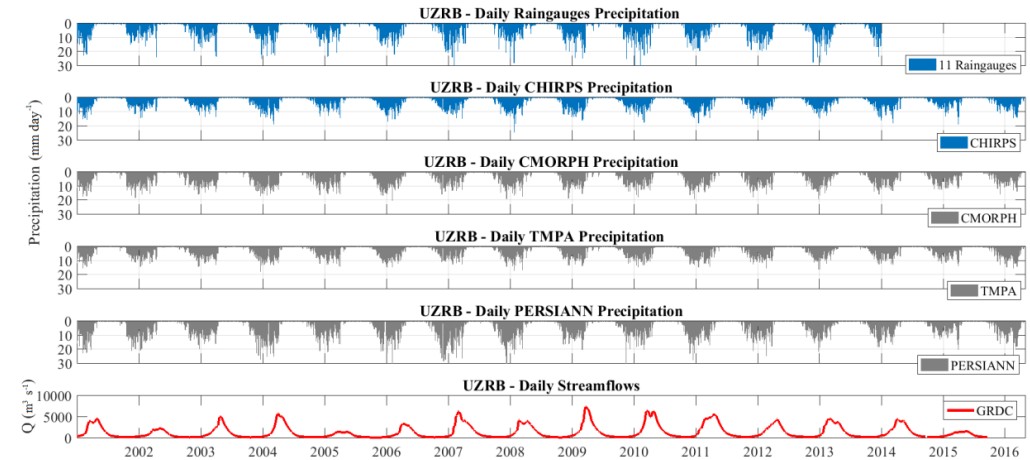

**Figure 4. Historical daily rainfall estimates (2001-2016) spatially-averaged for 11 rain gauges and SPPs (CMORPH, TMPA, and PERSIANN) in the UZRB. Daily streamflows at Katima Mulilo stream gauge are also presented (red line).**





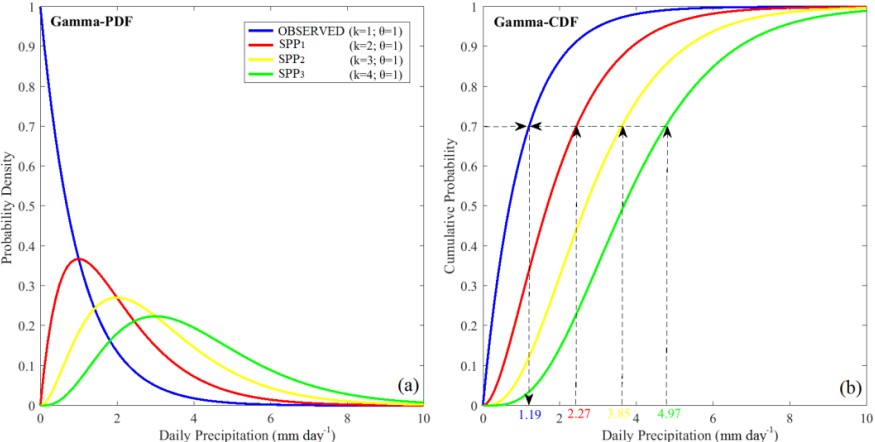

**Figure 5. Demonstration of the Quantile Mapping bias correction applied to daily precipitation estimates in the Upper Zambezi. (a) The Gamma Probability Density Function (Gamma-PDF) is hypothetically exemplified for observed CHIRPS and SPPs assuming different shapes (k parameter) for each dataset. (b) The respective Gamma Cumulative Distribution Function (Gamma-CDF) for CHIRPS and SPPs is matched for a Probability (P=0.7) using the Inverse Gamma Function, which is finally used to calculate the bias-corrected daily satellite precipitation estimates.**

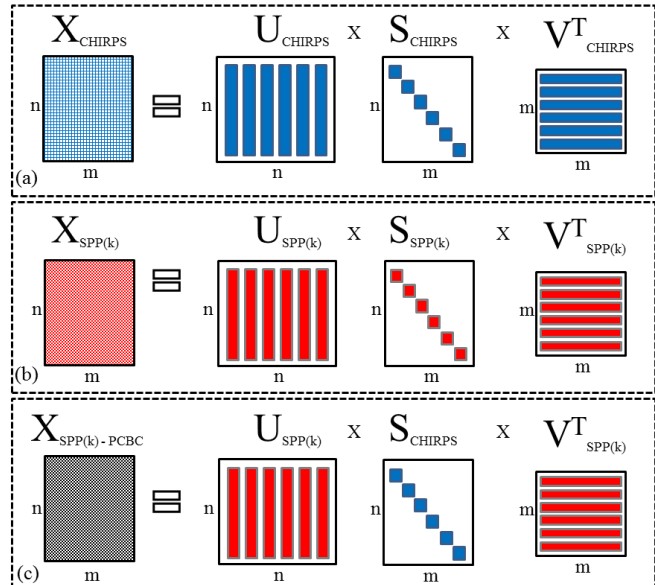

**Figure 6. Conceptual representation of (a) CHIRPS matrix ($X_{CHIRPS}$) decomposed as singular vectors ($U_{CHIRPS}$ and $V_{CHIRPS}$ and singular values $S_{CHIRPS}$); (b) Raw SPP matrix decomposed as singular vectors ($U_{SPP(k)}$ and $V_{SPP(k)}$) and singular values ($S_{SPP(k)}$); and (c) Bias-corrected SPP matrix reconstructed using the singular vectors calculated from SPPj combined with singular values calculated from observed CHIRPS data ($S_{CHIRPS}$).**





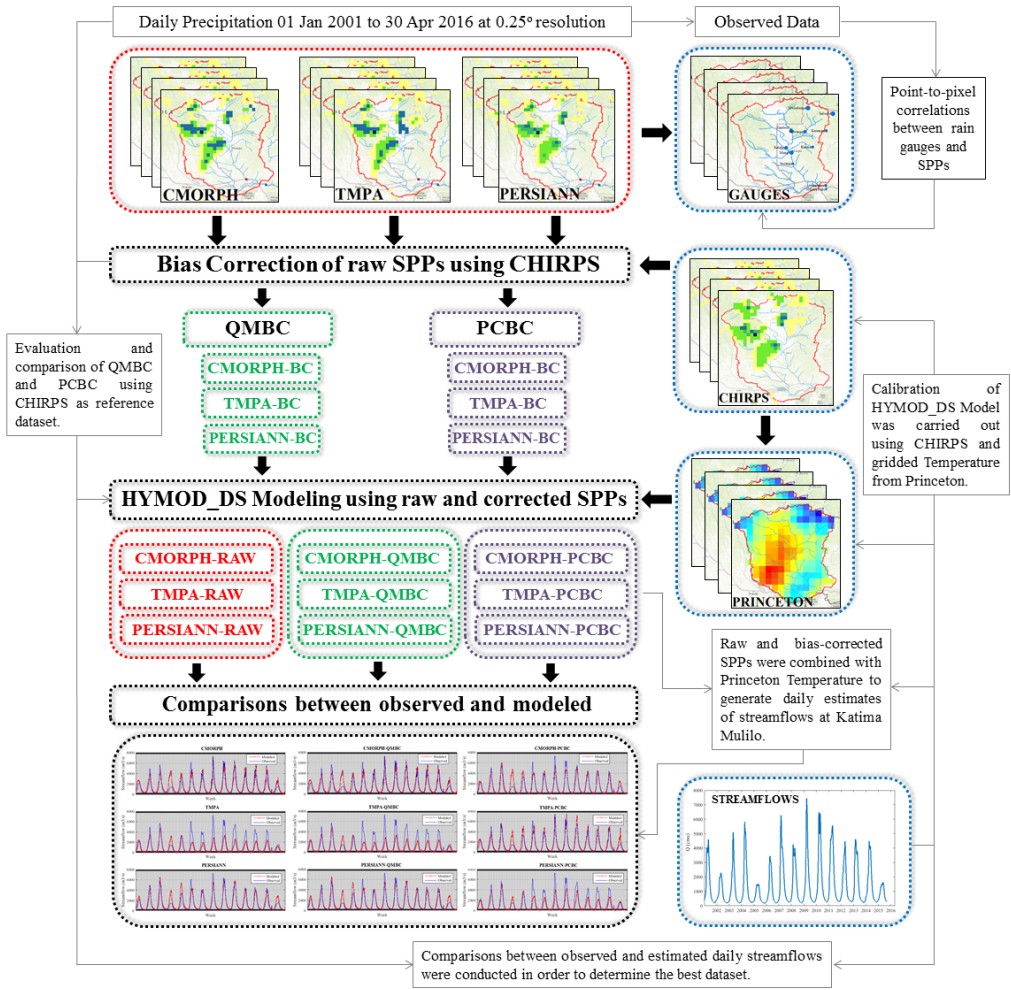

**Figure 7. Research structure used to evaluate the performance of BC methods for hydrologic forecasting in the UZRB.**





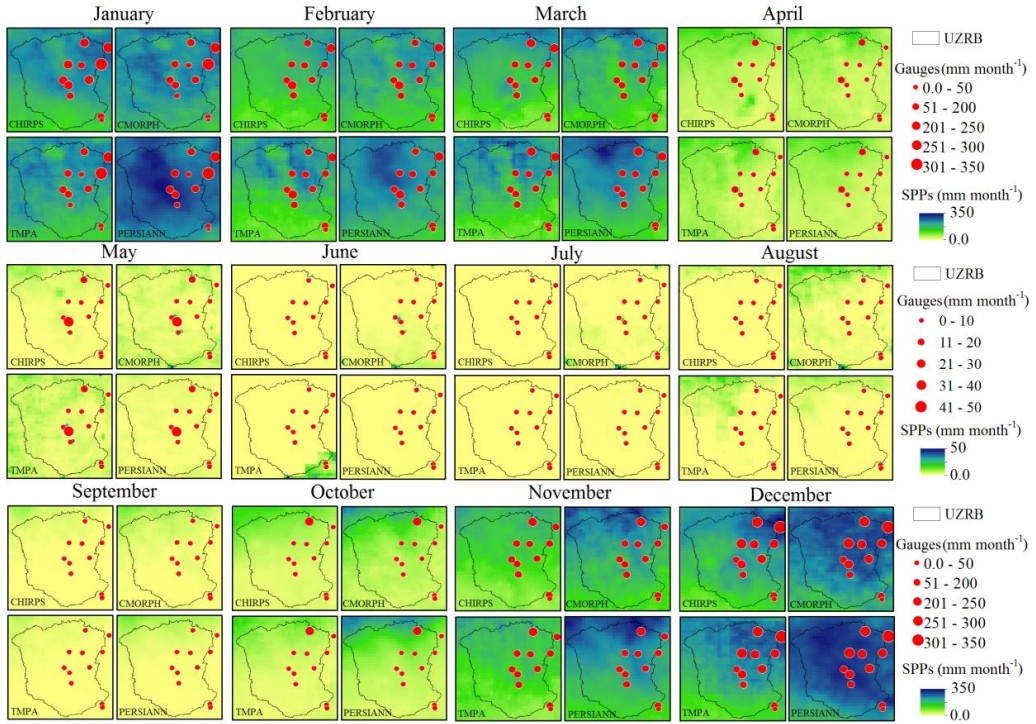

**Figure 8. Precipitation seasonality of the UZRB derived from mean monthly accumulations of CHIRPS, CMORPH, TMPA, and PERSIANN for the period 2001-2016. Monthly maps are superimposed with mean monthly precipitation accumulation recorded at 11 rain gauges (red dots) for the period 2001-2013.**





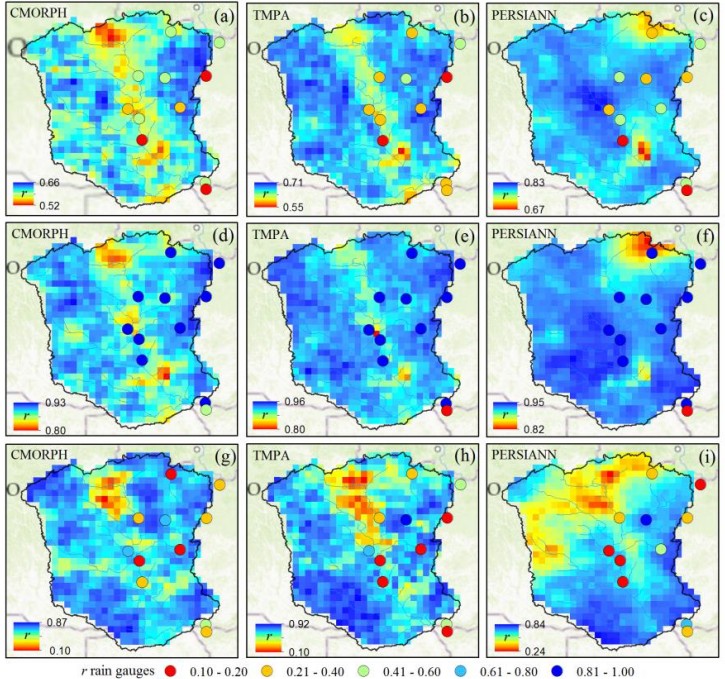

**Figure 9. Pixel-to-pixel correlations for daily (a),(b),(c), monthly (d),(e),(f), and annual (g),(h),(i) accumulations of raw CHIRPS versus raw SPPs (CMORPH, TMPA, and PERSIANN) for the period 01/01/2001 to 04/30/2016. Maps are superimposed with point-to-pixel correlations between daily, monthly, and annual precipitation accumulation recorded at 11 rain gauges versus SPPs. Illustrations (a),(d), and (g) are the correlations between CMORPH versus CHIRPS and rain gauges; illustrations (b),(e), and (h) are the correlations between TMPA versus CHIRPS and rain gauges; and illustrations (c), (f), and (i) are the correlations between PERSIANN versus CHIRPS and rain gauges.**

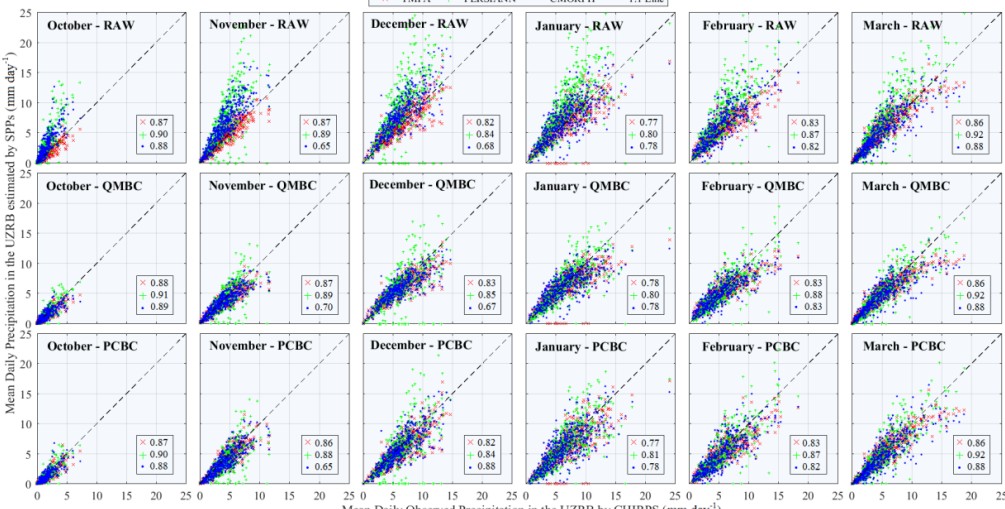

**Figure 10. Scatter plots for the relationship between daily raw and BC SPPs estimates (mm day⁻¹) versus CHIRPS. The correlations were performed for the rainy season (Oct-Mar) of the period 2001-2016. The correlation coefficient is included for each pair of estimates.**



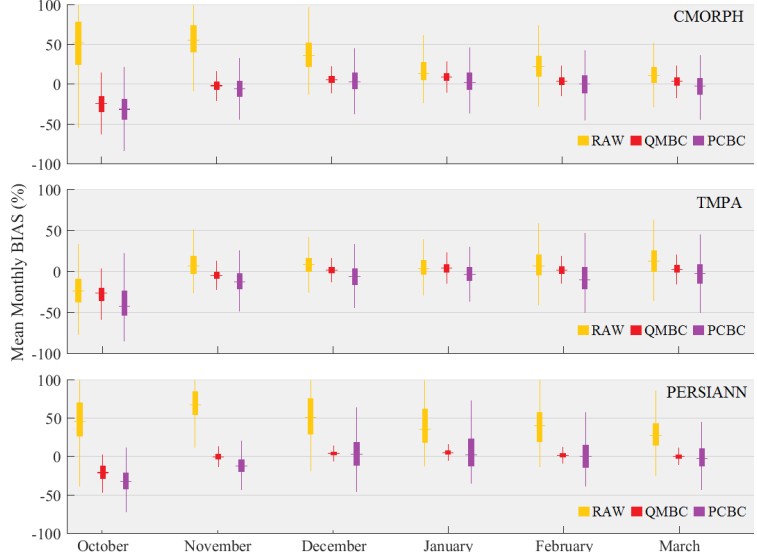

**Figure 11. Boxplots of monthly-grouped daily BIAS percentage for the wet season (Oct -Mar) of the period 2001-2016, calculated using raw and bias-corrected daily precipitation estimates from (a) CMORPH, (b) TMPA (TRMM 3B42-RT), and (c) PERSIANN.**

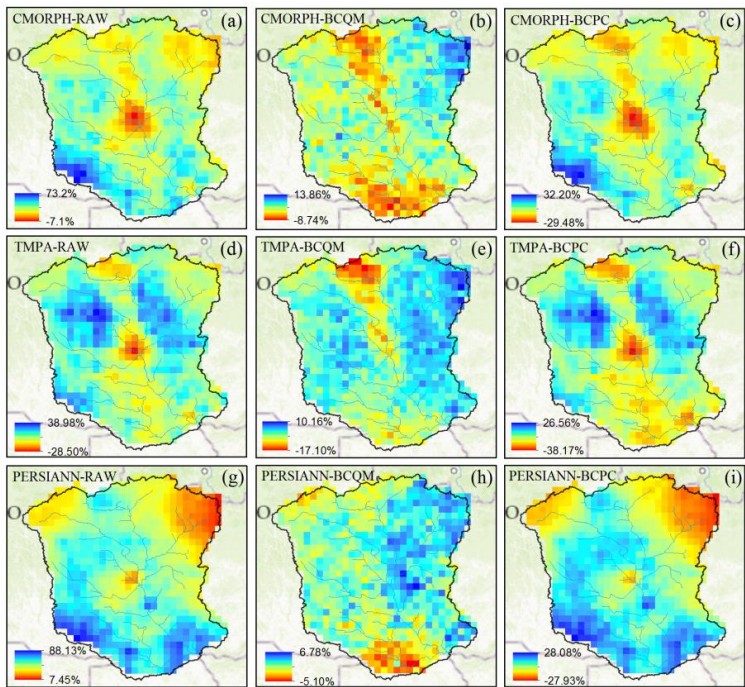

**Figure 12. Maps of mean daily BIAS Percentage (%) calculated for daily precipitation accumulation of CHIRPS versus raw and bias-corrected SPPs (CMORPH, TMPA, and PERSIANN) for the period 01/01/2001 to 04/30/2016. Illustrations (a),(b),(c) are the mean daily BIAS % calculated for CMORPH versus CHIRPS; illustrations (d),(e),(f) are the mean daily BIAS % calculated for TMPA versus CHIRPS; and illustrations (g),(h),(i) are the mean daily BIAS % calculated for PERSIANN versus CHIRPS.**




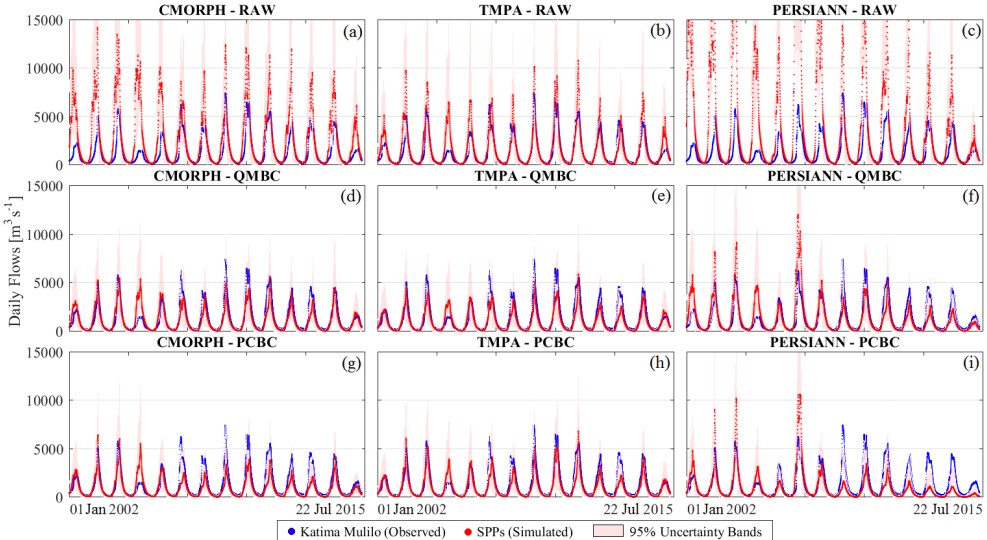

**Figure 13. Daily streamflow forecasts (m3 s$^{-1}$) at Katima Mulilo stream gauge using raw and bias-corrected SPPs.**

5    **Tables**

**Table 1.** UZRB rain gauges used in this study. For spatial reference see Fig. 1.

| Id | Name | South Latitude | West Longitude | Elevation (m.a.s.l.) | Period | Available Daily Records | Missing Daily Data (%) | Available Monthly Records | Missing Monthly Data (%) | Available Yearly Records | Missing Yearly Data (%) |
|---|---|---|---|---|---|---|---|---|---|---|---|
| 1 | Kalabo | -14.85 | 22.70 | 1033 | 2001-2011 | 3813 | 19.7 | 152 | 17.4 | 14 | 12.5 |
| 2 | Zambezi | -13.53 | 23.11 | 1076 | 2001-2013 | 4748 | 0.0 | 184 | 0.0 | 16 | 0.0 |
| 3 | Mongu | -15.25 | 23.15 | 1056 | 2001-2013 | 4718 | 0.6 | 183 | 0.5 | 16 | 0.0 |
| 4 | Senanga | -16.10 | 23.27 | 1001 | 2001-2012 | 1924 | 59.5 | 90 | 51.1 | 10 | 37.5 |
| 5 | Kabompo | -13.60 | 24.20 | 1107 | 2001-2005 | 1580 | 66.7 | 79 | 57.1 | 8 | 50.0 |
| 6 | Mwinilunga | -11.75 | 24.43 | 1329 | 2001-2013 | 4556 | 4.0 | 177 | 3.8 | 16 | 0.0 |
| 7 | Kaoma | -14.80 | 24.80 | 1155 | 2001-2013 | 4142 | 12.8 | 164 | 10.9 | 16 | 0.0 |
| 8 | Livingstone | -17.82 | 25.82 | 996 | 2001-2013 | 4748 | 0.0 | 184 | 0.0 | 16 | 0.0 |
| 9 | Kasempa | -13.53 | 25.85 | 1193 | 2001-2013 | 3050 | 35.8 | 127 | 31.0 | 11 | 31.3 |
| 10 | Victoria Falls | -18.10 | 25.85 | 1067 | 2001-2013 | 3813 | 19.7 | 147 | 20.1 | 13 | 18.8 |
| 11 | Solwezi | -12.18 | 26.38 | 1378 | 2001-2013 | 4748 | 0.0 | 183 | 0.5 | 16 | 0.0 |





**Table 2.** SPPs used in this study.

| Product Name | Temporal Resolution | Spatial Resolution (lat/lon) | Coverage | Period of Records | Main Reference | Description |
|---|---|---|---|---|---|---|
| CMORPH | 3-hourly | 0.25 x 0.25 | 60N - 60S | 01/01/2001 – 04/30/2016 | Joyce et al. (2004) | CMORPH rain rates are derived from Micro-Waves (MW) measurements, and geostationary InfraRed (IR) images are used to infer motion fields, which are then used to propagate the MW rain fields in space and time. CMORPH data is available at: http://ftp.cpc.ncep.noaa.gov/precip/global_CMORPH/ |
| TRMM 3B42-RT (TMPA) | 3-hourly | 0.25 x 0.25 | 50N - 50S | 01/01/2001 – 04/30/2016 | Huffman et al. (2007) | The Tropical Rainfall Measuring Mission (TRMM) Multisatellite Precipitation Analysis (TMPA) provides a calibration-based sequential scheme for combining precipitation estimates from multiple satellites. Data is available at: ftp://trmmopen.nascom.nasa.gov/pub/merged/mergeIRMicro/ |
| PERSIANN | 3-hourly | 0.25 x 0.25 | 60N - 60S | 01/01/2001 – 04/30/2016 | Sorooshian et al. (2014) | The PERSIANN algorithm fits the pixel brightness temperature and its neighbor temperature textures, in terms of means and standard deviations, to the calculated pixel rain rates based on an artificial neural network (ANN) model. PERSIANN data is available at: ftp://persiann.eng.uci.edu/pub/ |
| CHIRPS | Daily | 0.25 x 0.25 | 50N - 50S | 01/01/2001 – 04/30/2016 | Funk et al. (2015) | CHIRPS is a 30+ year quasi-global rainfall dataset. Starting in 1981 to near-present, CHIRPS incorporates 0.05° and 0.25° resolution satellite imagery with in-situ station data to create gridded rainfall time series for trend analysis and seasonal drought monitoring. Data is available at: ftp://ftp.chg.ucsb.edu/pub/org/chg/products/CHIRPS-2.0 |

**Table 3.** Comparison between daily observed flows (Katima Mulilo) and daily predicted flows obtained by using raw and bias-corrected SPPs estimates.

| Type | Product | Error Measures | | | | | | | | |
|---|---|---|---|---|---|---|---|---|---|---|
| | | NMSE | RMSE | NBE | NVE | NSE | r | PBE | AARE | PEMF |
| **Raw** | **CMORPH-RAW** | 2.70 | 2480.0 | 0.92 | 2.99 | -1.70 | 0.75 | 91.79 | 113.73 | 91.07 |
| | **TMPA-RAW** | 0.51 | 1072.8 | 0.22 | 0.83 | 0.49 | 0.87 | 21.97 | 43.78 | 45.58 |
| | **PERSIANN-RAW** | 14.23 | 5691.7 | 1.95 | 13.90 | -13.23 | 0.64 | 195.41 | 218.99 | 370.53 |
| **Quantile Mapping** | **CMORPH-QMBC** | 0.32 | 852.8 | -0.17 | -0.34 | 0.68 | 0.84 | -16.78 | 42.09 | -23.57 |
| | **TMPA-QMBC** | 0.30 | 821.8 | -0.26 | -0.50 | 0.70 | 0.89 | -25.75 | 39.65 | -20.65 |
| | **PERSIANN-QMBC** | 0.64 | 1203.1 | -0.04 | 0.34 | 0.36 | 0.74 | -4.41 | 52.97 | 61.93 |
| **Principal Components** | **CMORPH-PCBC** | 0.49 | 1052.3 | -0.33 | -0.53 | 0.51 | 0.78 | -32.85 | 47.24 | -13.49 |
| | **TMPA-PCBC** | 0.29 | 816.4 | -0.26 | -0.41 | 0.71 | 0.88 | -25.63 | 40.22 | -8.23 |
| | **PERSIANN-PCBC** | 0.77 | 1324.9 | -0.31 | 0.00 | 0.23 | 0.66 | -31.41 | 55.53 | 43.06 |

5    NMSE is the Normalized Mean Square Error; RMSE is the Root Mean Squared Error; NBE is the Normalized Bias Error; NVE is the Normalized Variance Error; NSE is the Nash-Sutcliffe Efficiency; r is the Correlation Coefficient; PBE is the Percentage Bias Error; AARE is the Average Absolute Relative Error; and PEMF is the Percentage Error in Maximum Flow.