# Peer review of "Bias correction of daily satellite-based rainfall estimates for hydrologic forecasting in the Upper Zambezi, Africa"

_Hydrology and Earth System Sciences, 2016_

## Referee Comment (RC1) · Anonymous Referee #1 · 20 Nov 2016

The manuscript by Valdés-Pineda et al presents an interesting application of satellite-based rainfall estimations feeding an hydrological model. The paper shows a logical structure and it is easy to read. The pipeline for such an application is appealing, but in my opinion, authors should improve the discussion of the results in order to provide better advise to readers interested on the topics, and to give a better context and motivation for the work done. Moreover, there are assumptions that are not clearly stated or supported by figures and text. Therefore, I suggest to present a more elaborated discussion of the results, before full acceptance.

The Introduction section would requiere a slight change on the wording. There is a key question in p2-l10 (page 2 - line 10) that is not further elaborated: Why there is a

need to perform site-specific and season-targeted bias correction?. Also, this section should present in more detail main factors affecting the estimation's bias (see p9-l1 to l4). Moreover, it is not clear if authors seek to test the magnitude of bias, the temporal persistence of bias, the spatial pattern of bias or simply the most efficient empirical method for a specific application (see p13-l5 to l9). The analysis of previous work on UZRB should include some discussion regarding the spatial and temporal resolution of the satellite-derived products against the observational scale (gauges) and processes scale (e.g. predominant rainfall and runoff processes). In my opinion, the transboundary nature of the UZRB is a relevant issue or driver to approach the use on satellite-derived information on water resources management. I suggest to include this analysis into the Introduction instead of the Study Area section. Please, indicate when maximum flows occur (p3-l35). Section 2.2 would requiere some supporting references for the SST-rainfall (observational) relationship. Also, I suggest to include and analysis terms of inter-annual to decadal variability. Most of the results and conclusions rely on assuming that the CHIRPS data set properly represent the patio-temporal patterns in the UZRB. However, the manuscript discusses this issue mainly in qualitative terms. I suggest to better present a quantitative assessment of the representativeness of the data set. Regarding the bias-correction methods, I would like to comment three issues. The first one is the potential influence of offsetting the drizzle effect to 1 mm. Is there any relationship between local rainfall intensity features and the 1 mm threshold? I would like to suggest a brief sensitivity analysis for this issue. The second issue is the assumption of the Gamma-PDF as the best surrogate for rainfall statistics. Authors should provide a quantitative assessment in a pixel-basis for the goodness of fit between empirical and observed distributions. The third issue is about the novel approach presented. It would be useful for readers to also include some analysis in terms of results of the eigenvectors and eigenvalues. For instance, are there significant changes on loads depending on the validation and calibration periods? . The use of an hydrological model to assess the performance of gridded or satellite-derived data is appealing. However, there are a few issues that should be discussed. First,

how authors are able to separate different uncertainty sources (input and structural)? There must be a discussion regarding the (potential) magnitude of model's uncertainty against input's uncertainty. Also, there is a lack of discussion regarding the ability of the model to properly represent the hydrological process within the basin (not only the streamflow time series). The Results section is mainly descriptive. I recommend to include more discussion. For example, p10-l33 states that a given results is anticipated for "all scales". However, the manuscript only shows daily and monthly values. Authors should rephrase these section or perform analyses at finer temporal scales (14-days, 14-days windowing). Also, please provide (plausible) explanations for the spatial patterns of estimates and bias. Are high/low values only related to elevation? How cover could affect estimates? Is bias relate to synoptic types (p12:l21)? I suggest to rewrite the Conclusion section. Currently, the authors include several conditional sentences instead of proven facts or result-supported comments. Authors should be more concise and precise on answering two or three research questions. I would be informative if along with Fig 8, authors present and compare estimates for dry and wet 3-days (or 1-week) composites. Thus, readers could compare estimation at finer time scales. Figures 4, 13 and 10 should be redrawn to improve its readability. Figure 8 could be presented in terms of differences, too. Figures 3 and 11 should follow the same format. I suggest to use on maps quartile (or other division) for the color scale in order to better identify spatial patterns.

p3-l25: use lower case for km. p4-l12: I suggest to delete lines 12 to 16, as the authors state well-known knowledge. p8-l19: The acronym EOF is not defined. Through the manuscript, authors use the terms "forecasted" and "simulated" as interchangeable terms. Please, be consistent. I would prefer the use of simulated. p11-l31: Lines 31-35 should be places as comments at the end of the manuscript as they not provide facts or conclusion supported by results. p13-l16: delete "always"

---

## Referee Comment (RC2) · Anonymous Referee #2 · 11 Dec 2016

This study compares three satellite-based precipitation products adjusted by two bias correction methods and evaluates performance of streamflow modeling forced by these products. This manuscript is a well-written case study for a data-sparse catchment where satellite precipitation information can play an important role to improve real-time hydrologic forecasting. However, throughout the manuscript, it was difficult to find a novel contribution or a new finding. A newly developed bias correction method, PCBC, lacks description on detailed procedures and advantages and could not demonstrate its improved performance over the conventional approaches in the most comparative results. Although the authors argued inclusion of additional components would improve the performance of PCBC, demonstration of superiority of a new algorithm is not a kind

of work which can be left as a future endeavor. In addition, applications and analysis on hydrologic forecasting lack essential components required for forecasting and do not provide improved understanding. Therefore, the manuscript is not recommended to be published in a high ranked journal, HESS. Despite this objection, if this manuscript would be accepted, I hope the followings would be addressed before final publication:

1. Detailed description, justification and demonstration of a new bias correction algorithm, PCBC:

- What are the advantages of PCBC over the conventional bias correction methods? Please elaborate the limitations of the conventional methods and how PCBC could overcome these limitations. In addition, please describe what advantages can be expected using this method from statistical and computational perspectives.

- Authors argued that performance of PCBC could be improved if additional components would be included. As mentioned above, this demonstration could not be left as a future research because the current results do not prove advances of the proposed methodology.

2. Limitation of PCBC:

- SPPs are crucial information for hydrologic forecasting in poorly gauged or ungauged basins (PUB). However, PCBC requires grid-based statistics on observation, which could make applications of this method for PUB inefficient or nearly impossible.

- More importantly, there is an unresolved question about whether adaptation of principal component without using the main benefit, reduction of the dimensionality, can be statistically useful to correct biases in precipitation information. As shown Figs. 12 and 13, PCBC failed to not only correct spatial pattern of bias in the raw data (Fig. 12) but also reduce the variance of bias (Fig. 13). The current version of PCBC seems to work only for reducing total sum of bias without significant improvement in spatial pattern and variance.

3. Hydrologic forecasting or retrospective modeling:

- The methodology used in this study can be used for a part of hydrologic forecasting, but lacks important other steps in hydrologic forecasting. Since satellite precip products are information for the current time step, without addressing and demonstrating the methodology using forecasted forcings, the current work is about not hydrologic forecasting, but hindcasting using historical data. If the manuscript could be meaningful in terms of hydrologic forecasting, the following research questions should be addressed and demonstrated: What precipitation and weather forcing could be used in the forecasting step without losing consistency to satellite precip info in the current time step? What sorts of bias correction would be used to adjust forecasted forcing having different spatio-temporal biases with varying lead times?

Specific comments:

4. Fig. 12: The range of legend should be the same among different sub-plots for the fare comparison. This rule should be applied for all figures comparing spatial distribution.

5. Many potential readers wonder how distribution of principal components and singular values in Eq. (5) look like. Please add one example in the appendix if available.

6. Fig. 13: Why do hydrologic simulations by PCBC show significant underestimation in the several flooding seasons?

7. Figs. 3 and 4 may not be required because observations are being presented in the other plots.
* * *

---

## Author Comment (AC1) · 9 Jan 2017

Dear Editor, first of all we would like to thank both reviewers for their comments and suggestions to our submission. The revised version of our manuscript (based on their comments) will definitely improve the quality and readability of our paper. The new manuscript will include most of the changes suggested by both referees. Detailed responses are attached below each reviewer's suggestion or comment using red font color. Thanks once again for your feedback and prompt response. Anonymous Referee #1 The manuscript by Valdés-Pineda et al presents an interesting application of satellite based rainfall estimations feeding an hydrological model. The paper shows a logical structure and it is easy to read. The pipeline for such an application is appeal-

ing, but in my opinion, authors should improve the discussion of the results in order to provide better advise to readers interested on the topics, and to give a better context and motivation for the work done. Moreover, there are assumptions that are not clearly stated or supported by figures and text. Therefore, I suggest to present a more elaborated discussion of the results, before full acceptance. We acknowledge this referee for his initial comments to our manuscript. We agree in the fact that our paper shows a logical structure and it is easy to read, but of course, there is still a chance of presenting a more elaborated discussion of the results, before full acceptance. Introduction Section The introduction section would requiere a slight change on the wording. There is a key question in p2-l10 (page 2 - line 10) that is not further elaborated: Why there is a need to perform site-specific and season-targeted bias correction?. Also, this section should present in more detail main factors affecting the estimation's bias (see p9-l1 to l4). In the new version of our manuscript we are including a modified version of the introduction section. The key questions stated at the end of this section will be answered and further elaborated. In general, there is a need to perform site-specific correction because every catchment is a unit with specific landscape and climatic features, and both of these can affect the performance of SPPs. The evaluation of season-targeted bias is also important and required given that the performance of SPPs is influenced by the way in which the sensors capture the information and how the algorithms use that information to estimate rainfall events i.e., SPPs cannot adequately differentiate between stratiform and convective rainfall events.

Moreover, it is not clear if authors seek to test the magnitude of bias, the temporal persistence of bias, the spatial pattern of bias or simply the most efficient empirical method for a specific application (see p13-l5 to l9). This is a great point of view. We agree that this is the most confusing part of our paper because we are including all these analyses in the current version of our paper (1. test the magnitude of bias, 2. Test the temporal persistence of bias, 3. Test the spatial pattern of bias, 4. Test the most efficient empirical method for a specific application). We will clarify what is the real motivation of our work in the new version of our manuscript. The analysis

of previous work on UZRB should include some discussion regarding the spatial and temporal resolution of the satellite-derived products against the observational scale (gauges) and processes scale (e.g. predominant rainfall and runoff processes). In my opinion, the transboundary nature of the UZRB is a relevant issue or driver to approach the use on satellite-derived information on water resources management. I suggest to include this analysis into the Introduction instead of the Study Area section. Please, indicate when maximum flows occur (p3-l35). We will move the paragraph including the transboundary nature of the UZRB from the Study Area Section to the Introduction Section. We will also include some discussion about the spatial and temporal resolution of the satellite-derived products against the observational scale (gauges) and processes scale (e.g. predominant rainfall and runoff processes)

Section 2.2 Section 2.2 would require some supporting references for the SST-rainfall (observational) relationship. Also, I suggest to include and analysis terms of inter-annual to decadal variability. We will expand Section 2.2. to describe in more detail the climatology of the Upper Zambezi River Basin. For these purposes we will improve the discussion about the sea surface temperature (SST) patterns and their resulting atmospheric teleconnections. We will also describe how these oceanic changes are related to inter-annual or decadal variability of rainfall in the basin. Most of the results and conclusions rely on assuming that the CHIRPS data set properly represent the patio-temporal patterns in the UZRB. However, the manuscript discusses this issue mainly in qualitative terms. I suggest to better present a quantitative assessment of the representativeness of the dataset. We agree that this is an important question that should be clarified in the manuscript. In fact, a quantitative analysis to compare the performance of CHIRPS and SPPs was carried out before submitting our paper (point to pixel correlations and hydrological simulations). This analysis revealed that CHIRPS climatology achieves better results than SPPs (in both correlations and simulations); therefore it can be truly used as reference dataset. In this regard we do not feel that this analysis should be included in the manuscript since our interest is not comparing climatology versus SPPs, but analyzing the performance of SPPs in the basin. Instead, we propose to include some paragraphs detailing the results of our quantitative assessment of CHIRPS and SPPs.

Bias Correction Methods Regarding the bias-correction methods, I would like to comment three issues. The first one is the potential influence of offsetting the drizzle effect to 1 mm. Is there any relationship between local rainfall intensity features and the 1 mm threshold? I would like to suggest a brief sensitivity analysis for this issue. This is a great comment. We carried out this analysis (before submitting our manuscript) by testing different thresholds between 0.1 and 2 mm. We found that these different thresholds do not have a significant influence in the correction of SPPs. We selected the threshold of 1 mm because it is the value more suggested by previous literature. We will mention this in the revised version of our manuscript. The second issue is the assumption of the Gamma-PDF as the best surrogate for rainfall statistics. Authors should provide a quantitative assessment in a pixel-basis for the goodness of fit between empirical and observed distributions. During our screening analysis we tested and compared different PDFs and finally determined that Gamma PDF (used for Quantile Mapping Method) is the best surrogate for rainfall statistics in the Upper Zambezi River Basin. This finding is strongly supported by previous literature. We will include more discussion about these results in the revised version of our manuscript. The third issue is about the novel approach presented. It would be useful for readers to also include some analysis in terms of results of the eigenvectors and eigenvalues. For instance, are there significant changes on loads depending on the validation and calibration periods? Regarding this comment it is important to mention that both the Quantile Mapping and the Principal Components corrections are applied using all historical records which are updated every day with new data (climatology and SPPs); therefore, a comparison between validation and calibration periods is not required (used) for the real-time forecasting. Instead, will include an analysis and a figure presenting the results of the eigenvalues and eigenvectors to provide more details to the readers about our results.

[Figure]

Section 2.7 The use of an hydrological model to assess the performance of gridded or satellite-derived data is appealing. However, there are a few issues that should be discussed. First, how authors are able to separate different uncertainty sources (input and structural)? There must be a discussion regarding the (potential) magnitude of model's uncertainty against input's uncertainty. Also, there is a lack of discussion regarding the ability of the model to properly represent the hydrological process within the basin (not only the streamflow time series).

This is another great observation from this referee. We did not include a discussion (or comparison) about the magnitude of input and structural errors since the objective of our manuscript was to exclusively analyze the extent of input errors rather than structural erors. For instance, we compared the bias from different satellite products (input uncertainty) assuming that the input errors can appropriately be quantified if all other sources of uncertainty are kept constant. For example, we used the same model states and conditions for all simulations we performed. In this way, we were able to focus our analyses in the propagation of errors from the input sources rather than the influence of model's structure. We will include more details about the experimental design carried out in our study plus some explanations about why we did not modify the structure of the model. Results The Results section is mainly descriptive. I recommend to include more discussion. For example, p10-l33 states that a given results is anticipated for "all scales". However, the manuscript only shows daily and monthly values. Authors should rephrase these section or perform analyses at finer temporal scales (14-days, 14-days windowing). Also, please provide (plausible) explanations for the spatial patterns of estimates and bias. Are high/low values only related to elevation? How cover could affect estimates? Is bias relate to synoptic types (p12:l21)? We analyzed daily, monthly and yearly scales, which is the main reason of including ''all scales'' in the wording of p10-l33. The section will be rephrased including these new suggestions. The spatial pattern of estimates and bias will be related to elevation features in the basin as way to reveal plausible explanations for the spatial patterns of bias in the basin. Conclusions I suggest to rewrite the Conclusion section. Currently, the au-

[Figure]

thors include several conditional sentences instead of proven facts or result-supported comments. Authors should be more concise and precise on answering two or three research questions. This section will be rewritten according with modifications done in the revised version of our manuscript. Figures I would be informative if along with Fig 8, authors present and compare estimates for dry and wet 3-days (or 1-week) composites. Thus, readers could compare estimation at finer time scales. Since Figure 8 presents the performance of SPPs along the seasonality of the basin, we feel that adding finer scales will not contribute in giving better insights about our results. It is also not clear how the composites suggested by this referee should be grouped and how many months should be included in the analysis.

Figures 4, 13 and 10 should be redrawn to improve its readability. Figure 8 could be presented in terms of differences, too. Figures 3 and 11 should follow the same format. I suggest to use on maps quartile (or other division) for the color scale in order to better identify spatial patterns. Accordingly with these suggestions, we will include updated figures in the revised version of our manuscript. We tried different classification schemes for the maps and concluded that the one used currently in the paper is the best one for comparison purposes.

p3-l25: use lower case for km. p4-l12: I suggest to delete lines 12 to 16, as the authors state well-known knowledge. p8-l19: The acronym EOF is not defined. Through the manuscript, authors use the terms "forecasted" and "simulated" as interchangeable terms. Please, be consistent. I would prefer the use of simulated. p11-l31: Lines 31-35 should be places as comments at the end of the manuscript as they not provide facts or conclusion supported by results. p13-l16: delete "always" We will include all these suggestions in the revised version of our manuscript.

---

## Author Comment (AC2) · 9 Jan 2017

Dear Editor, this referee has made valuable contributions to improve our manuscript. Most of them had been also stated by referee 1, therefore we have shortened our responses in this revision. In general we will be adopting most of the suggestions and comments proposed by this referee trying to keep the consistency between both revisions. This referee refers mostly to improvements that can be applied to our novel PCBC method. Most of these suggestions are easy to handle, however we will appreciate if you can also give us some feedback about the best way to present our manuscript to the readers. Thanks once again for your comments and suggestions to our submission. This study compares three satellite-based precipitation products

adjusted by two bias correction methods and evaluates performance of streamflow modeling forced by these products. This manuscript is a well-written case study for a data-sparse catchment where satellite precipitation information can play an important role to improve real-time hydrologic forecasting. However, throughout the manuscript, it was difficult to find a novel contribution or a new finding. A newly developed bias correction method, PCBC, lacks description on detailed procedures and advantages and could not demonstrate its improved performance over the conventional approaches in the most comparative results. Although the authors argued inclusion of additional components would improve the performance of PCBC, demonstration of superiority of a new algorithm is not a kind of work which can be left as a future endeavor. In addition, applications and analysis on hydrologic forecasting lack essential components required for forecasting and do not provide improved understanding. Therefore, the manuscript is not recommended to be published in a high ranked journal, HESS. Despite this objection, if this manuscript would be accepted, I hope the followings would be addressed before final publication: We acknowledge this author for pointing out the fact of our novel contribution for bias correction. As mentioned in the previous review (referee 1) we did not clarify our motivation (or objective) adequately; therefore, there is still an opportunity to describe and demonstrate the potential of PCBC in correcting SPPs. At some point we thought that it would be more valuable having a new and more elaborated version of PCBC in a new paper; however, given the recommendations and suggestions of this reviewer we will include them in the revised version of our manuscript.

1. Detailed description, justification and demonstration of a new bias correction algorithm, PCBC: - What are the advantages of PCBC over the conventional bias correction methods? Please elaborate the limitations of the conventional methods and how PCBC could overcome these limitations. In addition, please describe what advantages can be expected using this method from statistical and computational perspectives. - Authors argued that performance of PCBC could be improved if additional components would be included. As mentioned above, this demonstration could not be left as a future

research because the current results do not prove advances of the proposed methodology. We will elaborate a more detailed description, justification and demonstration of PCBC, mentioning the advantages and disadvantages over conventional methods. We will also mention the computational advantages of PCBC and will include an example about how the retention of a less number of components could improve or reduce the performance of this method.

2. Limitation of PCBC: - SPPs are crucial information for hydrologic forecasting in poorly gauged or ungauged basins (PUB). However, PCBC requires grid-based statistics on observation, which could make applications of this method for PUB ineficient or nearly impossible. - More importantly, there is an unresolved question about whether adaptation of principal component without using the main benefit, reduction of the dimensionality, can be statistically useful to correct biases in precipitation information. As shown Figs. 12 and 13, PCBC failed to not only correct spatial pattern of bias in the raw data (Fig. 12) but also reduce the variance of bias (Fig. 13). The current version of PCBC seems to work only for reducing total sum of bias without significant improvement in spatial pattern and variance. We will provide the results of PCBC including an example of how the reduction of the dimensionality could potentially benefit the bias correction of SPPs. We understand that all these methods (Quantile Mapping and PCBC) are limited in poorly gauges basins; however, we also know that the trend of hydrological applications is migrating towards the implementation of new products, especially gridded datasets from remote sensing. The revised version of our manuscript will include these new results.

3. Hydrologic forecasting or retrospective modeling: - The methodology used in this study can be used for a part of hydrologic forecasting, but lacks important other steps in hydrologic forecasting. Since satellite precip products are information for the current time step, without addressing and demonstrating the methodology using forecasted forcings, the current work is about not hydrologic forecasting, but hindcasting using historical data. If the manuscript could be meaningful in terms of hydrologic forecasting, the following research questions should be addressed and demonstrated: What precipitation and weather forcing could be used in the forecasting step without losing consistency to satellite precip info in the current time step? What sorts of bias correction would be used to adjust forecasted forcing having different spatio-temporal biases with varying lead times? Specificomments: 4. Fig. 12: The range of legend should be the same among different sub-plots for the fare comparison. This rule should be applied for all figures comparing spatial distribution. 5. Many potential readers wonder how distribution of principal components and singular values in Eq. (5) look like. Please add one example in the appendix if available. 6. Fig. 13: Why do hydrologic simulations by PCBC show significant underestimation in the several flooding seasons? 7. Figs. 3 and 4 may not be required because observations are being presented in the other plots.

In this paragraph there are several questions and comments that were previously stated by referee 1 and answered in the previous review. We will be including more discussion and results about PCBC method dealing with this referee's comments. We will include the distribution of leading modes from PCBC and will modify all figures accordingly with his suggestions.